# Prototype Based Classification from Hierarchy to Fairness

## Abstract

Artificial neural nets can represent and classify many types of high-dimensional data but are often tailored to particular applications – e.g., for "fair" or "hierarchical" classification. Once an architecture has been selected, it is often difficult for humans to adjust models for a new task; for example, a hierarchical classifier cannot be easily transformed into a fair classifier that shields a protected field. Our contribution in this work is a new neural network architecture, the concept subspace network (CSN), which generalizes existing specialized classifiers to produce a unified model capable of learning a spectrum of multi-concept relationships. We demonstrate that CSNs reproduce state-of-the-art results in fair classification when enforcing concept independence, may be transformed into hierarchical classifiers, or may even reconcile fairness and hierarchy within a single classifier. The CSN is inspired by and matches the performance of existing prototype-based classifiers that promote interpretability.

## 1 Introduction

Neural networks are able to learn rich representations of data that support highly accurate classification; however, understanding or controlling what neural nets learn remains challenging. Some techniques offer insight into pre-trained models by uncovering directions within latent spaces that correspond to particular concepts, image manipulations, or more (Goetschalckx et al., 2019; Kim et al., 2018), while approaches focused on interpretability provide techniques that are more comprehensible to humans (Li et al., 2018; Chen et al., 2019). While these methods provide insight, they fail to offer control: humans observe learned patterns but are unable to guide models such that learned relationships are useful for a particular setting or task.

Another line of work has advanced the design of models for particular types of classification tasks (such as fair or hierarchical classification) but these techniques are often developed with only one problem in mind (Zemel et al., 2016; Xie et al., 2017; Hase et al., 2019). For example, models built for fair classification (predicting an outcome regardless of information about a protected field) are only used to enforce independence of concepts rather than hierarchy. Thus, humans may exert control over learned representations by selecting an appropriate technique rather than tuning training parameters within the same technique.

We have designed a new neural network architecture, the concept subspace network (CSN), which generalizes existing specialized classifiers to produce a unified model capable of learning a spectrum of multi-concept relationships. CSNs use prototype-based representations, a technique employed in interpretable neural networks in prior art (Li et al., 2018; Chen et al., 2019; Garnot & Landrieu, 2020). A single CSN uses sets of prototypes in order to simultaneously learn multiple concepts; classification within a single concept (e.g., "type of animal") is performed by projecting encodings into a concept subspace defined by the prototypes for that concept (e.g., "bird," "dog," etc.). Lastly, CSNs use a measure of concept subspace alignment to guide concept relationships such as independence or hierarchy.

In our experiments, CSNs performed comparably to state-of-the art in fair classification, despite prior methods only being designed for this type of problem. In applying CSNs to hierarchical classification tasks, networks automatically deduced interpretable representations of the hierarchical problem structure, allowing them to outperform state-of-the-art, for a given neural network backbone, in terms of both accuracy and average cost of errors on the CIFAR100 dataset. Lastly, in

a human-motion prediction task, we demonstrated how a single CSN could enforce both fairness (to preserve participant privacy) and hierarchy (to exploit a known taxonomy of tasks). Our findings suggest that CSNs may be applied to a wide range of problems that had previously only been addressed individually, or not at all.

## 2 RELATED WORK

### 2.1 INTERPRETABILITY AND PROTOTYPE NETWORKS

Numerous post-hoc explanation techniques fit models to pre-trained neural nets; if humans understand these auxiliary models, they can hypothesize about how the neural nets behave (Ribeiro et al., 2016; Lundberg & Lee, 2017). However, techniques in which explanations are decoupled from underlying logic may be susceptible to adversarial attacks or produce misleading explanations (Heo et al., 2019; Slack et al., 2020).

Unlike such decoupled explanations, interpretability research seeks to expose a model's reasoning. In this work we focus on prototype-based latent representations in neural nets. There is a long history of learning discrete representations in continuous spaces, originating under "vector quantization" literature (Kohonen, 1990; Schneider et al., 2009). More recently, the prototype case network (PCN) comprised an autoencoder model that clustered encodings around understandable, trainable prototypes, with classifications made via a linear weighting of the distances from encodings to prototypes (Li et al., 2018). Further research in image classification extended PCNs to use convolutional filters as prototypes and for hierarchical classification in the hierarchical prototype network (HPN) (Chen et al., 2019; Hase et al., 2019). Lastly, Garnot & Landrieu (2020) use prototypes in Metric-Guided Prototype Learning (MGP) in conjunction with a loss function to cluster prototypes to minimize user-defined costs.

Our model similarly uses trainable prototypes for classification, but differs from prior art in two respects. First, we modify the standard PCN architecture to support other changes, without degrading classification performance. Second, like HPNs (but not PCNs or MGP), CSNs leverage multiple sets of prototypes to enable hierarchical classification but also allow for non-hierarchical concept relationships.

### 2.2 FAIR AND HIERARCHICAL CLASSIFICATION

AI fairness research considers how to mitigate undesirable patterns or biases in machine learning models. Consider the problem of predicting a person's credit risk: non-causal correlations between age and risk may lead AI models to inappropriately penalize people according to their age (Zemel et al., 2016). The problem of fair classification is often framed as follows: given inputs, $x$, which are informative of a protected field, $s$, and outcome, $y$, predict $y$ from $x$ without being influenced by $s$ (Zemel et al., 2013). Merely removing $s$ from $x$ (e.g., not including age as an input to a credit predictor) rarely removes all information about $s$, so researchers have developed a variety of techniques to create representations that "purge" information about $s$ (Zemel et al., 2016; Xie et al., 2017; Jiang et al., 2020).

Hierarchical classification solves a different problem: given a hierarchical taxonomy of classes (e.g., birds vs. dogs at a high level and sparrows vs. toucans at a low level), output the correct label at each classification level. Neural nets using convolution and recurrent layers in specialized designs have achieved remarkable success in hierarchical image classification (Zhu & Bain, 2017; Guo et al., 2018). The hierarchical prototype network (HPN) uses prototypes and a training routine based upon conditional subsets of training data to create hierarchically-organized prototypes (Hase et al., 2019). Garnot & Landrieu (2020) also use prototypes for hierarchical classification in Metric-Guided Prototype Learning (MGP) by adjusting the training loss to guide prototype arrangement. Neither HPN nor MGP explicitly models relationships between multiple subsets of prototypes. Lastly, recent works propose hyperbolic latent spaces as a natural way to model hierarchical data (Dai et al., 2021; Mathieu et al., 2019; Nickel & Kiela, 2017; Liu et al., 2020). Our method, conversely, relies upon concepts from Euclidean geometry. Extending the principle of subspace alignment that we develop to non-Euclidean geometric spaces is a promising direction but is beyond the scope of this work.

## 3 TECHNICAL APPROACH

In this section, we outlined the design of the CSN, which was inspired by desires for both interpretable representations and explicit concept relationships. First, we wished for interpretable representations, so we built upon the PCN design, with modifications. Second, we explicitly encoded relationships between concepts by introducing multiple sets of prototypes, instead of just one in PCNs. Third, we enabled guidance of the concept relationships by modifying the CSN training loss. Together, these changes supported not only interpretable classification, but also provided a flexible framework for a single model architecture to learn different concept relationships.

### 3.1 CONCEPT SUBSPACE CLASSIFICATION

A CSN performing a single classification task (e.g., identifying a digit in an image) is defined by three sets of trainable weights. First, an encoder parametrized by weights $\theta$, $e_\theta$, maps from inputs of dimension $X$ to encodings of dimension $Z$: $e_\theta : R^X \to R^Z$. Second, a decoder parametrized by weights $\phi$, $d_\phi$, performs the decoding function of mapping from encodings to reconstructed inputs: $d_\phi : R^Z \to R^X$. Third, there exist a set of $k$ trainable prototype weights, $\boldsymbol{p}$, that are each $Z$-dimensional vectors: $p_1, p_2, ..., p_k \in R^Z$. This architecture resembles that of the PCN, but without the additional linear classification layer (Li et al., 2018).

Here, we focus briefly on the set of prototypes, $\boldsymbol{p}$. Given a set of $k$ prototypes in $R^Z$, we define a "concept subspace," $C$ as follows:

$$v_i = p_i - p_1 \quad \forall i \in [2, k] \tag{1}$$

$$C = \{x | x \in R^Z \text{ where } x = p_1 + \sum_{i \in [2,k]} \lambda_i v_i \text{ for } \lambda_i \in R \quad \forall i\} \tag{2}$$

$C$ is the linear subspace in $R^Z$ defined by starting at the first prototype and adding linear scalings of vector differences to all other prototypes. We call this subspace a concept subspace because it represents a space of encodings between prototypes defining a single concept (e.g., prototypes for digits 0, 1, 2, etc. define a concept subspace for digit classification).

A CSN's architecture — consisting of an encoder, a decoder, and a set of prototypes and the associated concept subspace — enables two types of functionality: the encoder and decoder may be composed to reconstruct inputs via their latent representations, and CSNs may perform classification tasks by mapping an input, $x$, to one of $Y$ discrete categories. Classification is performed by first encoding an input into a latent representation, $z = e_\theta(x)$. The $l2$ distance from $z$ to each prototype is then calculated, yielding $k$ distance values: $d_i(z, \boldsymbol{p}) = ||z - p_i||_2^2; i \in [1, k]$. These distances are mapped to a probability distribution, $\mathbb{P}_K(i); i \in [1, k]$, by taking the softmax of their negatives. Lastly, if there are more prototypes than classes, (e.g., two prototypes for dogs, two for cats, etc.) the distribution over $k$ is converted to a distribution over $Y$ categories summing the probabilities for prototypes belonging to the same class.

For single-concept classification, CSNs differ from PCNs primarily by removing the linear layer that PCNs used to transform distances to prototypes into classifications. We found this unnecessary for high classification accuracy (Appendix A) and instead directly used negative distances. Without the linear layer, CSN classification is equivalent to projecting encodings, $z$, onto a concept subspace before calculating distances. The distances between projected encoding, dubbed $z_{proj}$, and prototypes will induce the same softmax distribution as when the orthogonal component remains. Indeed, we find projection more intuitive - only the component of $z$ that corresponds to the subspace is used for classification - and list projection as a standard step in the remainder of this paper. A simple example of projecting an encoding and calculating distances to prototypes is shown in Figure 1 a.

For some tasks, we used an encoder design from variational-autoencoders (VAEs) in order to regularize the distribution of encodings to conform to unit Gaussians (Kingma & Welling, 2014). By default, this regularization loss was set to 0, but it sometimes proved useful in some domains to prevent overfitting (as detailed in experiments later). We emphasize that CSNs are discriminative, rather than generative, models, so we did not seek to learn a latent space from which to sample.

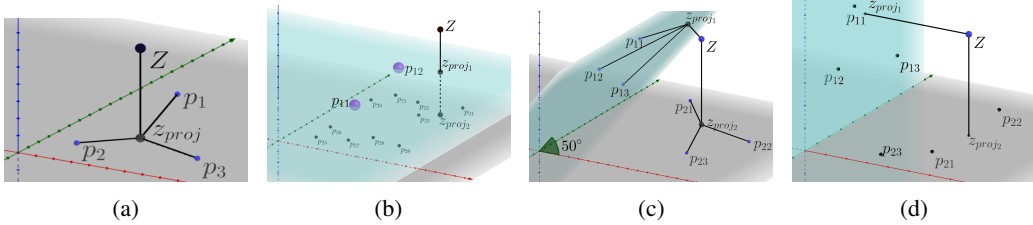

Figure 1: CSNs support single (a) or multiple (b-d) classification by projecting encodings into concept subspaces. Subspaces may exhibit a range of relationships from parallel (b) to orthogonal (d).

## 3.2 MULTI-CONCEPT LEARNING

We defined the CSN architecture for single classification tasks in the previous section; here, we explain how a CSN may be used for multiple classification tasks. (For example, consider a scenario involving classifying both what type of bird a photo depicts and whether the photo was taken outdoors or indoors.) Extending CSNs to support multiple classifications requires the addition of new sets of prototypes. This is the primary contribution of our work.

Multiple classification tasks are performed by defining a set consisting of sets of prototypes: $\mathbb{P} = \{\boldsymbol{p_1}, ..., \boldsymbol{p_c}\}$, with a set of prototypes for each of $c$ classification tasks. A classification task is performed by using the CSN's encoder to generate an encoding, $z$, and projecting $z$ into the concept subspace defined by the set of prototypes particular to the given task. Figure 1 (b-d) depicts simplified examples of two concept subspaces. In each example, each concept uses three prototypes, yielding two planar concept spaces (one of which corresponds to the $x - y$ plane for illustrative purposes); $z$ may be projected into either plane depending upon the classification task at hand.

While the prototypes in different sets are separate from each other, correlations present in training data may lead to a range of relationships among prototypes. Returning to the previous example scenario, prototypes of birds may represent canaries and toucans, while prototypes of indoor and outdoor scenes may represent living rooms and jungles; each set of prototypes is independent in principle, but in reality, prototypes may represent canaries in living rooms and toucans in jungles. In fact, two sets of prototypes can exhibit a range of relationships from highly correlated to fully independent, as shown in Figure 1.

We defined a metric, concept subspace *alignment*, to reflect this range of relationships. Mathematically, the alignment of two subspaces is the mean of the cosine squared of the angle between all pairs of vectors drawn from the basis of each subspace. Given orthonormal bases, efficiently computed via $QR$ factorization, $Q_1$ and $Q_2$, of ranks $m$ and $n$, we define alignment as follows:

$$a(Q_1, Q_2) = \frac{1}{mn} \sum_{i}^{m} \sum_{j}^{n} (Q_1^\top Q_2[i, j])^2 \tag{3}$$

Given the range of values for the cosine squared function, alignment values range from 0 to 1 for orthogonal and parallel subspaces, respectively. Intuitively, orthogonality lends itself well to independent concepts and therefore supports fair classification, whereas parallel subspaces naturally correspond to hierarchical classification. We elaborated on this intuition in Section 3.4.

## 3.3 TRAINING PROCEDURE

When training a CSN, we assume access to a set of training data, $(X, \mathbb{Y})$ for $\mathbb{Y} = (Y_1, Y_2, ...Y_c)$. For each entry in the dataset, there is an input $x$ and a label $y_i$, for each of $c$ classification tasks.

We trained CSNs in an end-to-end manner to minimize a single loss function, defined in Equation 4. The four terms in the loss function were as follows: 1) reconstruction error; 2) the loss introduced for the PCN, encouraging classification accuracy and the clustering of encodings around prototypes (applied within each concept subspace); 3) a KL divergence regularization term; and 4) a term penalizing alignment between concept subspaces. Each term was weighted by a choice of real-

valued $\lambda$s. We emphasize that the PCN loss — clustering and classification accuracy, defined in Equation 7 of Li et al. (2018) — is calculated within each concept subspace using the projections of encodings; thus, encodings were encouraged to cluster around prototypes only along dimensions within the subspace. The encoder, decoder, and prototype weights were trained simultaneously.

$$
\begin{aligned}
l(X, \mathbb{Y}, \theta, \phi, \mathbb{P}) = & \frac{\lambda_0}{|X|} \sum_{x \in X} (d_\phi(e_\theta(x)) - x)^2 \\
& + \sum_{i \in [1,C]} \lambda_{Pi} PCN(\texttt{proj}(e_\theta(X), \boldsymbol{p_i}), Y_i) \\
& + \sum_{i \in [1,C]} \lambda_{KLi} KL(X, \boldsymbol{p_i}) \\
& + \sum_{i \in [1,C]} \sum_{j \in [1,C]} \lambda_{Aij} a(Q_i, Q_j)
\end{aligned}
\tag{4}
$$

The KL regularization term mimics training losses often used in VAEs that penalize the divergence between the distribution of encodings and a zero-mean unit Gaussian (Kingma & Welling, 2014). In our case, we wished to induce a similar distribution of encodings, but centered around prototypes rather than the origin. Furthermore, rather than induce a Gaussian distribution within a concept subspace (which would dictate classification probabilities and therefore potentially worsen classification accuracy), we wished to regularize the out-of-subspace components of encodings.

Concretely, we implemented this regularization loss in three steps. First, we computed the orthogonal component of an encoding as $z_{orth} = z - z_{proj}$. We then computed the KL divergence between the distribution of $z_{orth}$ and unit Gaussians centered at each prototype in each subspace. Finally, we took the softmax over distances between encodings and prototypes in order to only select the closest prototype to the encoding; we then multiplied the softmax by the divergences to enforce that encodings were distributed as unit Gaussians around the nearest prototype in each subspace. Together, these operations led the distributions out of each subspace to conform to unit Gaussians around each prototype. As confirmed in later experiments, this component was crucial in training fair classifiers.

### 3.4 Hierarchical and Fair Classification

We conclude this section by demonstrating how CSNs may support hierarchical or fair classifications. Hierarchical and fair classification may be thought of as extremes along a spectrum of concept alignment. In hierarchical classification, concepts are highly aligned and therefore parallel: the difference between a toucan and a Dalmatian is similar to the difference between a generic bird and dog, and so the vector differences between prototypes associated with different classes should also be parallel (e.g., "bird" - "dog" = "toucan" - "Dalmatian."). In fair classification, concepts are not aligned: switching belief about someone's sex should not alter predictions about their income. Thus, based on the classification task, moving an encoding relative to one subspace should either affect (for hierarchical) or not affect (for fair) that encoding's projection onto the other subspace. We provide a geometric interpretation of these two tasks in Figure 1 b and d.

CSNs can be trained to adopt either form of concept relationship by penalizing or encouraging concept subspace alignment (already present as $a(Q_i, Q_j)$ in the training loss). Our single model reconciles these two types of problems by viewing them as opposite extremes along a spectrum of concept relationships that our technique is able to learn; this is the main contribution of our work.

## 4 Results

Our experiments were divided in four parts. First, we demonstrated how CSNs matched standard performance on single classification tasks: in other words, that using a CSN did not degrade performance. We omit these unsurprising results from the paper; full details are included in Appendix A. Second, we showed that CSNs matched state-of-the-art performance in two fair classification tasks. Third, we used CSNs for hierarchical classification tasks, exceeding performance demonstrated by

Table 1: Mean Adult dataset fairness results.

| Model | $y$ Acc. | $s$ Acc. | D.I. | DD-0.5 |
|---|---|---|---|---|
| CSN | 0.85 | 0.67 | 0.83 | 0.16 |
| Adv. | 0.85 | 0.67 | 0.87 | 0.16 |
| VFAE | 0.85 | 0.70 | 0.82 | 0.17 |
| FR Train | 0.85 | 0.67 | 0.83 | 0.16 |
| Wass. DB | 0.81 | 0.67 | 0.92 | 0.08 |
| Random | 0.76 | 0.67 | | |

Table 2: Mean German dataset fairness results.

| Model | $y$ Acc. | $s$ Acc. | D.I. | DD-0.5 |
|---|---|---|---|---|
| CSN | 0.73 | 0.81 | 0.70 | 0.10 |
| Adv. | 0.73 | 0.81 | 0.63 | 0.10 |
| VFAE | 0.72 | 0.81 | 0.47 | 0.23 |
| FR Train | 0.72 | 0.80 | 0.55 | 0.16 |
| Wass DB | 0.72 | 0.81 | 0.33 | 0.02 |
| Random | 0.70 | 0.81 | | |

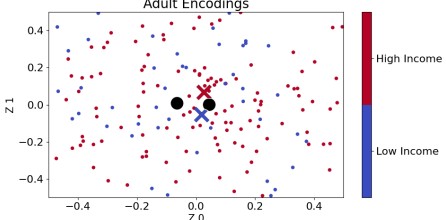

Figure 2: A 2D latent space for fair classification. The prototypes for high and low income (X) are perpendicular to the prototypes for applicant sex (circles).

Table 3: Fairness ablation results on the German dataset. Alignment enforced causal independence, and additionally including the KL loss enforced distributional independence. All std. err. $< 0.02$.

| Loss | $y$ Acc. | D.I. | DD-0.5 | $\rho$ |
|---|---|---|---|---|
| KL + Align | 0.73 | **0.70** | **0.10** | 0.00 |
| Align | 0.73 | 0.69 | 0.11 | 0.00 |
| KL | 0.72 | 0.67 | 0.12 | 0.20 |
| Neither | **0.75** | 0.64 | 0.13 | 0.13 |

prior art along several metrics. Fourth, we showed how CSNs enabled both fair and hierarchical classification in a dataset describing human motion in an assembly task that exploited hierarchical knowledge while preserving participant anonymity. Implementation details of CSNs in all experiments are included in Appendix F.

## 4.1 FAIR CLASSIFICATION

We evaluated CSN's performance in fair classification tasks in the Adult and German datasets. These datasets are commonly used in fairness literature and contain data that can be used to predict people's income or credit risks (Dua & Graff, 2017).We compared CSN performance to our implementations of an advesarial purging technique (Adv.), the variational fair autoencoder (VFAE), Wasserstein Fair Classification (Wass. DB), and a mutual-information-based fairness approach (FR Train) (Xie et al., 2017; Zemel et al., 2016; Jiang et al., 2020; Roh et al., 2020). Implementation details of fair classification baselines and full results including standard deviations are included in Appendix G.

For the Adult dataset, the protected attribute was sex, and for the German dataset, the protected attribute was a binary variable indicating whether the person was older than 25 years of age. In evaluation, we measured $y$ Acc., the accuracy of predicting income or credit, $s$ Acc., the accuracy of a linear classifier trained to predict the protected field from the latent space, disparate impact (DI), as defined in Roh et al. (2020), and demographic disparity (DD-0.5), as defined by Jiang et al. (2020).

Mean results over 20 trials for both datasets were included in Tables 1 and 2. In both datasets, we observed that CSNs matched state-of-the-art performance. CSNs produced high $y$ Acc., indicating high task performance for predicting income or credit. Furthermore, fairness measures demonstrate that CSNs purged protected information successfully (low $s$ Acc.) and achieved high D.I. and low DD-0.5, as desired. A visualization of the latent space of a fair classifier, trained on the German dataset, is shown in Figure 2 and confirmed that CSNs learned orthogonal concept subspaces.

In addition to reproducing the state of the art, we conducted an ablation study to demonstrate the importance of two terms in our training loss: the alignment and KL losses. Using the German dataset, we trained 20 CSNs, setting the KL, alignment, or both loss weights to 0. The mean results of these trials are reported in Table 3.

Table 3 demonstrates the necessity of both KL and alignment losses to train fair predictors (with higher disparate impact and lower demographic disparity values). Including both loss terms resulted

in the fairest predictors; removing those losses could enable better classification accuracy, but at the expense of fairness. This confirms geometric intuition: the alignment loss created orthogonal subspaces and the KL regularization created distributional equivalence based on the subspaces. Jointly, these losses therefore produced statistical independence.

Table 3 also includes causal analysis of trained CSNs via the $\rho$ metric. Intuitively, this metric reflected the learned correlation between $s$ and $y$; it was calculated by updating embeddings in the CSN latent space along the gradient of $s$ and recording the change in prediction over $y$. We reported the ratio of these changes as $\rho$; as expected, enforcing orthogonality via alignment loss led to $\rho$ values of 0. This technique is inspired by work in causally probing language models (e.g., Tucker et al. (2021)); full details for calculating $\rho$ are included in Appendix D.

## 4.2 HIERARCHICAL CLASSIFICATION

We compared CSNs to our implementation of HPNs and results for Metric-Guided Prototype Learning (MGP), reported by Garnot & Landrieu (2020), for hierarchical classification tasks. Our HPN baseline used the same architecture as CSN (same encoder, decoder, and number of prototypes). It differed from CSNs by setting alignment losses to 0 and by adopting the conditional probability training loss introduced by Hase et al. (2019). We further included results of a randomly-initialized CSN under "Init." in tables. In these experiments, we sought to test the hypothesis that CSNs with highly aligned subspaces would support hierarchical classification, just as orthogonal subspaces enabled fair classification.

In addition to standard accuracy metrics, we measured two aspects of CSNs trained on hierarchical tasks. First, we recorded the "average cost" (AC) of errors. AC is defined as the mean distance between the predicted and true label in a graph of the hierarchical taxonomy (e.g., if true and predicted label shared a common parent, the cost was 2; if the common ancestor was two levels up, the cost was 4, etc.) (Garnot & Landrieu, 2020). Second, we measured the quality of trees derived from the learned prototypes. After a CSN was trained, we defined a fully-connected graph $\boldsymbol{G} = (\boldsymbol{V}, \boldsymbol{E})$ with vertices $\boldsymbol{V} = \mathbb{P} \bigcup \{\boldsymbol{0}\}$ (the set of all prototypes and a point at the origin) and undirected edges between each node with lengths equal to the $l2$ distance between nodes in the latent space. We recovered the minimum spanning tree, $\boldsymbol{T}$, from $\boldsymbol{G}$, (which is unique given distinct edge lengths, which we observed in all experiments), and converted all edges to directed edges through a global ordering of nodes. Lastly, we calculated the graph edit distance (ED) between isomorphisms of the recovered tree and the ground-truth hierarchical tree (with edges that obeyed the same ordering constraints) (Abu-Aisheh et al., 2015). Intuitively, this corresponded to counting how many edges had to be deleted or added to the minimum spanning tree to match the taxonomy tree, ignoring edge lengths, with a minimum value of 0 for perfect matches.

As a basic test of CSNs in hierarchical classification tasks, we created simple hierarchies from the MNIST Digit and Fashion datasets. The Digit dataset used the standard low-level labels of digit, supplemented with high-level labels of parity (two classes); the Fashion dataset used the standard low-level labels for item of clothing, with a ternary label for a high-level classification of "tops" (t-shirts, pullovers, coats, and shirts), "shoes" (sandals, sneakers, and ankle boots), or "other" (trousers, dresses, and bags).

Mean results from 10 trials for both MNIST datasets were included in Tables 4 and 5. The HPN baselines were implemented using the same number of prototypes as the CSNs being compared against. Both tables show that CSNs exhibit comparable or better accuracy than HPNs for both the low-level ($Y_0$) and high-level ($Y_1$) classification tasks. In addition, the average cost (A.C.) and edit distance (E.D.) values show that CSNs recovered minimum spanning trees that nearly perfectly matched the ground truth tree, and that when CSNs did make errors, they were less "costly" than errors made by HPNs (although admittedly, a dominant force in A.C. is classification accuracy alone). A 2D visualization of the latent space of a CSN trained on the Digit task is shown in Figure 3: encodings for particular digits clustered around prototypes for those digits (X), while prototypes for even and odd digits (circles) separated the digit clusters into the left and right halves of the latent space. Visualizations of latent spaces for more fair and hierarchical classification tasks are included in Appendix C; they confirmed the theoretical derivations of orthogonal and parallel subspaces.

Lastly, we trained 10 CSNs and HPNs on the substantially more challenging CIFAR100 dataset. The dataset is inherently hierarchical: the 100 low-level classes are grouped into 20 higher-level

Table 4: MNIST digit hierarchy mean (stdev) over 10 trials. First two columns × 100.

Table 5: MNIST fashion hierarchy mean (stdev) over 10 trials. First two columns × 100.

| | $Y_0\%$ | $Y_1\%$ | A.C. | E.D. | | $Y_0\%$ | $Y_1\%$ | A.C. | E.D. |
|---|---|---|---|---|---|---|---|---|---|
| CSN | 98 (0) | 99 (0) | 0.06 (0.00) | 3.4 (4.3) | CSN | 88 (0) | 98 (0) | 0.28 (0.01) | 0.0 (0.0) |
| HPN | 96 (0) | 98 (0) | 0.12 (0.01) | 23 (1.0) | HPN | 88 (0) | 98 (0) | 0.28 (0.01) | 24 (0.9) |
| Init. | 10 (0) | 50 (1) | 2.80 (0.01) | 20 (0.0) | Init. | 10 (0) | 33 (1) | 3.13 (0.02) | 30 (0.8) |

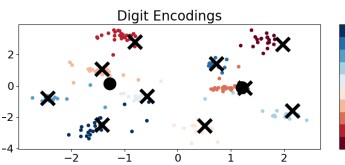

Figure 3: 2D latent space for hierarchical digit classification creates clusters around even and odd prototypes (circles on the right and left, respectively) and digit prototypes (X).

Table 6: CIFAR100 hierarchy results using the standard two-level hierarchy (top half) or MGP's 5-level hierarchy (bottom half).

| | $Y_0\%$ | $Y_1\%$ | A.C. | E.D. |
|---|---|---|---|---|
| CSN | **0.76** (0.0) | 0.85 (0.0) | **0.76** (0.02) | 11.2 (7) |
| HPN | 0.71 (0.0) | 0.80 (0.0) | 0.97 (0.04) | 165.0 (3) |
| Init. | 0.01 | 0.05 | 3.88 | 200 |
| CSN | **0.78** (0.0) | 0.88 (0.0) | **0.91** (0.0) | 6.0 (8.2) |
| MGP | 0.76 | – | 1.05 | – |
| Init. | 0.01 | 0.05 | 7.33 | 258 |

classes, each of size 5. Using a resnet18 encoder, pre-trained on ImageNet, in conjunction with 100 prototypes for low-level classification and 20 for high-level, we trained CSNs and HPNs. CSNs additionally used an alignment loss weight of -10 to encourage parallelism between the two concept subspaces. The mean results over 10 trials are shown in the top half of Table 6.

We also compared CSNs to MGP and other hierarchical classifiers using the CIFAR100 dataset and a deeper hierarchy, consisting of 5 levels of sizes 100, 20, 8, 4, and 2, as done by Garnot & Landrieu (2020). The additional information provided by this deeper hierarchy resulted in improved classification performance. Median results (as done by Garnot & Landrieu (2020)) for 10 CSNs using this dataset are shown in the bottom half of Table 6. Changing the hierarchy changed how average cost was calculated, so values from the top and bottom halves of the table should not be compared. Within the bottom half, we note that CSNs outperformed MGP on both A.C. and classification accuracy. Furthermore, according to values generated in the extensive experiments conducted by Garnot & Landrieu (2020), CSNs outperformed numerous other baselines, including HXE and soft-labels (Bertinetto et al. (2020)), YOLO (Redmon et al. (2016)), and a hyperspherical prototype network (Mettes et al. (2019)), all of which were built upon a resnet18 pretrained on ImageNet. In fact, our CSNs achieve SOTA classification accuracy for any classifier built upon a resnet18 backbone, without data augmentation. Furthermore, the decrease in A.C. is especially surprising given that other techniques explicitly optimized for average cost reductions, while CSNs merely trained on classification at each level. Notably, the decrease in A.C. is not fully explained by the increase in accuracy, indicating that CSN not only exhibited higher accuracy but also, when it did make mistakes, those mistakes were less severe.

Lastly, we note that CSNs support a range of learned relationships other than fair or hierarchical. The varying values of $\rho$ in Table 3 indicate that CSNs may learn different relationships when alignment loss is set to 0. However, in general, one could train models to learn desired relationships by penalizing or rewarding alignment relative to some intercept. We trained and evaluated such models in Appendix E and found that models indeed learned the desired alignment.

### 4.3 FAIR AND HIERARCHICAL CLASSIFICATION

Prior experiments demonstrated how CSNs could solve different classification problems separately; in this section, we applied a single CSN to a task that required it to use both fair *and* hierarchical classification. Intuitively, fairness was used to protect privacy, while hierarchical structure was used for better performance.

We used a dataset describing human motion in a bolt-placement task. The dataset was gathered from a similar setup to (Lasota et al., 2014) - motion was recorded at 50 Hz, using the 3D location of each of the 8 volunteer participant's gloved right hands as they reached towards one of 8 holes arranged

Table 7: CSNs for fair and hierarchical classification on bolt task. Mean (std. dev.) over 10 trials.

| Fair | Hier | Bolt% | $\rho$ | A.C. | D.I. | DD-0.5 |
|------|------|-------|--------|------|------|--------|
| No | No | 0.81 (0.01) | 0.36 (0.07) | 2.7 (0.05) | 0.80 (0.02) | 0.13 (0.02) |
| Yes | No | 0.41 (0.07) | $10^{-3}(< 10^{-3})$ | 2.8 (0.10) | 0.93 (0.03) | 0.05 (0.02) |
| Yes | Yes | 0.55 (0.06) | $10^{-3}\ (< 10^{-3})$ | 2.4 (0.05) | 0.92 (0.02) | 0.06 (0.02) |

in a line to place a bolt in the hole. The bolt holes may be thought of hierarchically by dividing destinations into left vs. right (LR) groupings, in addition to the label of the specific hole.

Initial exploration of the dataset showed promising results for prototype-based classification: the target locations were identified with 81% accuracy, and further analysis showed that prototypes corresponded to human-like motions (details in Appendix B). Troublingly, however a trained CSN could identify the participant with over 60% accuracy, which posed privacy concerns. Nevertheless, from a robotic safety perspective, it is important for robots to exploit as much information as possible to avoid collisions with humans.

Ultimately, we wished to predict which hole a participant was reaching towards given the past 1 second of their motion, while preserving privacy and exploiting hierarchical structure. Thus, we designed a CSN with three concept subspaces: one for predicting the bolt (8 prototypes), one for predicting the high-level grouping of a left or right destination (2 prototypes) and one for predicting the participant id (8 prototypes). We enforced that the bolt and LR subspaces were parallel while the bolt and participant subpsaces were orthogonal.

Results from our experiments are presented in Table 7. Means and standard deviations over 10 trials for each row are reported. (A.C. was calculated only for prediction errors, due to the large differences in accuracy rates across models.) When training a CSN with no constraints on subspace alignment, we found a highly accurate but unfair predictor (81% accuracy for bolt location, but sub-optimal disparate impact and demographic disparity values). Switching the CSNs to be fair classifiers by only enforcing orthogonality between bolts and participants yielded a fair classifier (illustrated by $\rho$, D.I., and DD-0.5), but with much worse bolt prediction accuracy (44%). However, by using a hierarchical subspace for LR groupings, the final CSN both improved classification accuracy and decreased the average cost of errors, while maintaining desired fairness characteristics.

## 5 CONTRIBUTIONS

The primary contribution of this work is a new type of model, the Concept Subspace Network, that supports inter-concept relationships. CSNs' design, motivated by prior art in interpretable neural network models, use sets of prototypes to define concept subspaces in neural net latent spaces. The relationships between these subspaces may be controlled during training in order to guide desired model characteristics. Critically, we note that two popular classification problems — fair and hierarchical classification — are located at either end of a spectrum of concept relationships, allowing CSNs to solve each type of problem in a manner on par with techniques that had previously been designed to solve only one. Furthermore, a single CSN may exhibit multiple concept relationships, as demonstrated in a privacy-preserving hierarchical classification task.

While we have demonstrated the utility of CSNs within several domains, numerous extensions could improve their design. First, the idea of subspace alignment could be applied to non-Euclidean geometries like hyperbolic latent spaces that are sometimes used for hierarchical classification. Second, CSNs could additionally benefit from relaxation of some simplifying assumptions: notably, allowing for more complex relationships rather than those defined by subspace cosine similarity, or using adversarial approaches for distributional regularization rather than only supporting unit Gaussians.

Lastly, we note that CSNs, while designed with ethical applications such as fair classification in mind, may lead to undesired consequences. For example, malicious actors could enforce undesirable concept relationships, or simply observing emergent concept relationships within a CSN could reinforce undesirable correlations. In addition, although prototypes encourage interpretability, which we posit can be used for good, the reductive nature of prototypes may be problematic when classifying human-related data (e.g., the COMPAS fair classification task we avoided).

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

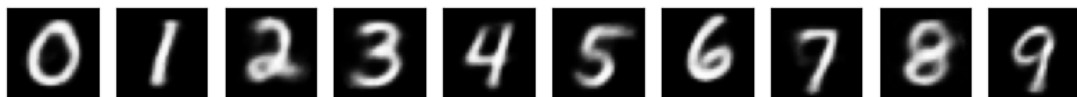

Figure 4: Decoded prototypes for digit classification provide visualizations of how images are classified.

Xinqi Zhu and Michael Bain. B-cnn: branch convolutional neural network for hierarchical classification. *arXiv preprint arXiv:1709.09890*, 2017.

## A SINGLE-CONCEPT CLASSIFICATION BASELINES

In addition to the specialized fair and hierarchical classification tasks, we tested CSNs on two standard classification tasks: identifying digits in the MNIST Digit dataset, and identifying one of 100 categories in the CIFAR100 dataset. There were no concept relationships present because this was a single classification task; instead, the tests established that using a CSN did not degrade classification performance relative to PCNs or other neural architectures.

On the Digit dataset, we trained a CSN using the same encoding and decoding layer architectures with 20 prototypes (two for each digit) and applied the same Gaussian distortions to training images as Li et al. (2018). Over five trials, training for 50 epochs with batches of size 250, we achieved the same mean classification accuracy as PCN (99.22%), demonstrating that the use of a CSN did not worsen classification accuracy (Li et al., 2018).

On the CIFAR100 dataset, we extended a resnet18 backbone (pretrained on ImageNet) as our encoder with 100 prototypes, 1 for each class, and trained 10 models for 60 epochs He et al. (2016). We achieved a mean classification accuracy of 76%, the standard result for networks built upon a resnet18 framework (Hase et al., 2019). Thus, CSNs exhibited high performance in a challenging domain, matching performance of normal networks, with the benefit of interpretable prototypes.

## B VISUALIZING DECODED PROTOTYPES

Because CSNs are built upon prototype-based classification, they are at least as interpretable as prior art, such as PCNs. In this section, we demonstrate how prototypes may be decoded to visualize their representations. These figures were generated by decoding prototypes from the models used throughout the paper.

Figure 4 shows the first 10 prototypes from the MNIST digit classifier in Appendix A. Unlike PCNs, CSNs benefit from an inductive bias that leads to an equal number of prototypes per class.

Figure 5 depicts the decoded prototypes when training a CSN to predict human reaching motions, as described in Section 3.4. This model was only trained on motion prediction, without using fair or hierarchical training terms. Interestingly, by over-parametrizing the number of prototypes (there were only 8 possible destinations, but twice as many prototypes), the model learned different forms of trajectories that reached towards the same destination: short movements near the targets, and longer loops when reaching from farther away.

## C VISUALIZING LEARNED LATENT SPACES

In addition to decoding prototypes, we visualized the latent spaces of trained classifiers. For the purposes of visualization, we trained new models from scratch, using only 2D latent spaces. Encodings and prototypes for both fair classification tasks, as well as the digit and fashion hierarchical classification tasks, are shown in Figure 6.

In all diagrams, encodings of test inputs are denoted by small colored dots. All classification tasks used 2 sets of prototypes: we depicted one set of prototypes as large black dots, and the other as X's. The arrangement of the prototypes in the latent spaces confirms that CSNs have learned the right concept alignment.

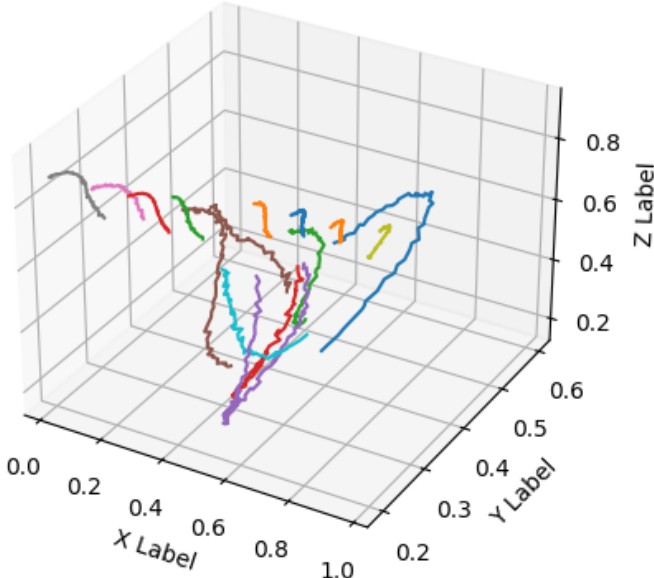

Figure 5: Decoded prototypes for the bolt classification task, plotted within an axis-normalized 3D space. Using 16 prototypes for the 8-label classification task reveals bolt-specific actions as well as a variety of motions such as screwing in the bolt and moving towards it.

Specifically, for the fair classification tasks, the X's form a line segment that is orthogonal to the line formed by the black dots. This orthogonality leads to fairness, as discussed in our paper.

In the hierarchical domains, we similarly observed that CSNs had learned the "right" latent structure. In these domains, the black dots denoted prototypes for high-level classification (such as even vs. odd). We observed that the lower-level prototypes (e.g., for digit), denoted by 'X's were clustered around the high level prototypes.

As a whole, these visualizations confirm that CSNs learn the desired latent structure, all controlled by changing the alignment loss weight.

## D    CALCULATING LEARNED CASUAL RELATIONSHIPS

In Sections 4.1 and 4.3, we report a metric, $\rho$, to denote the learned causal relationship between concepts. Here, we explain how we calculate $\rho$ in greater detail.

Intuitively, $\rho$ corresponds to the mean change in belief for one classification task divided by changes in belief for another classification task. For example, in fair classification, prediction of a person's credit risk should not change based on changes in belief over the person's age; this notion corresponds to $\rho = 0$.

We calculate $\rho$ in CSNs using a technique inspired by Tucker et al. (2021). In that work, the authors studied if a language model's output changed when the model's internal representation changed according to syntactic principles. In our work, we change latent representations, $z$, by taking the gradient of $z$ with respect to the loss of one classification task given true label $y^*$, creating a new $z'$

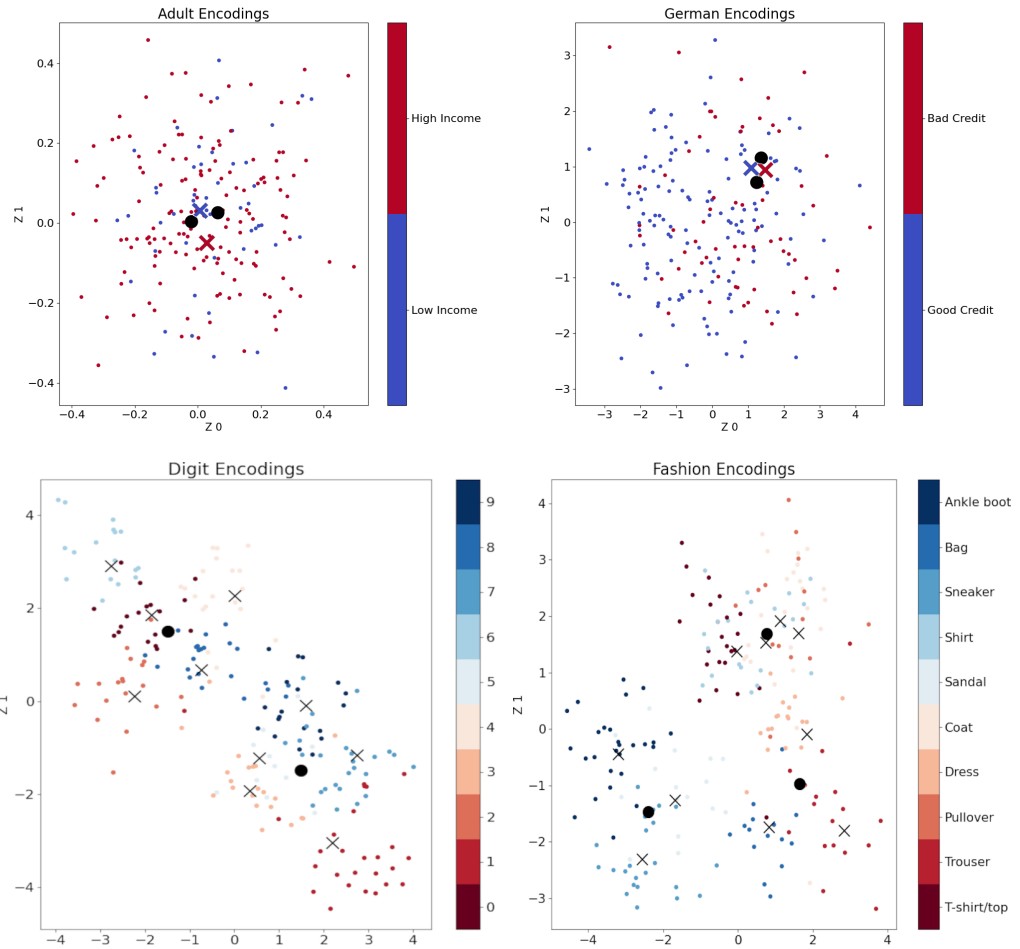

Figure 6: 2D learned latent spaces for the Adult, German, Fashion, and Digit datasets, showing prototypes from two concepts ($\times$'s and black dots). Fair tasks used orthogonal concept spaces (top); hierarchical tasks created multi-level clusters (bottom).

taken by moving $z$ along that gradient, and then calculating the new classification likelihoods using $z'$.

For simplicity, we limit our analysis to CSNs with two concept subspaces. We denote the encoding of an input, $x$, as $z = e_\theta(x)$. Prediction for each of the two tasks may be denoted as prediction functions, $\texttt{pred}_i$, for $i \in [0, 1]$ indicating the two tasks; the prediction function corresponds to projecting $z$ into the relevant subspace and calculating distances to prototypes, as discussed earlier. Using this notation, we define $\rho$ formally:

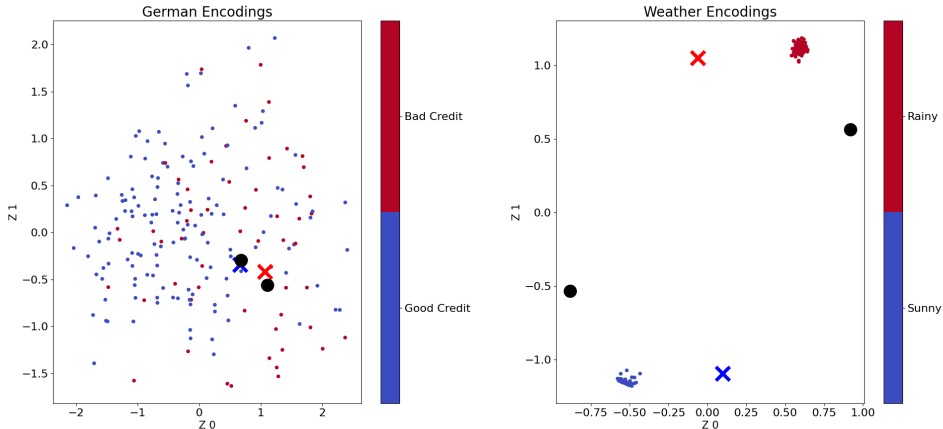

Figure 7: Without alignment loss guidance, CSNs may learn undesirable relationships between concepts like age (circles denote prototypes) and credit (Xs denote prototypes) in the German dataset (left). In an artificial weather scenario (right), CSNs can be guided to learn the right causal relationship between temperature and precipitation.

$$z = e_\theta(x) \tag{5}$$
$$y_0 = \texttt{pred}_0(z) \tag{6}$$
$$y_1 = \texttt{pred}_1(z) \tag{7}$$
$$z' = z + \nabla loss(y_0, y_0^*) \tag{8}$$
$$y_0' = \texttt{pred}_0(z') \tag{9}$$
$$y_1' = \texttt{pred}_1(z') \tag{10}$$
$$\rho = \frac{y_0 - y_0'}{y_1 - y_1'} \tag{11}$$

Thus, $\rho$ captures how the model's change in belief about one attribute affects its change in belief over another attribute, in other words the causal learned relationship between prediction tasks.

## E    CORRELATED-CONCEPT CLASSIFICATION BASELINES

A single CSN may be used to perform multiple classification tasks simultaneously without explicitly guiding concept relationships. In the Adult and German fair classification domains, CSNs predicted both $s$, the protected field, and $y$, the desired final prediction, while we explicitly guided the learned concept relationships to enforce fairness. In a separate set of experiments conducted on the same datasets, we demonstrated how CSNs can learn more complex concept relationships.

In these experiments, we trained CSNs with two subspaces, each with two prototypes, and set both the KL and alignment losses to zero. The CSNs were trained to predict both $s$ and $y$, using the two subspaces. We recorded prediction accuracy of $y$ and $\rho$, the learned causal correlation between $s$ and $y$.

Over 10 trials, for the German and Adult datasets, CSNs achieved mean $y$ classification accuracies of 85% and 74%, on par with prior art on these datasets when not enforcing fairness (Xie et al., 2017). We also found non-zero $\rho$: for the German dataset, we found a value of 0.20; for the Adult dataset, a mean value of 0.23. An example latent space from a CSN trained on the German dataset in this manner is shown in Figure 7, using the same visualization mechanism as introduced in Appendix C. In this example, the model learned a non-zero correlation between prototypes for credit (Xs) and

for an applicant's age (circles). This type of learned correlation is undesirable in fair classification domains but may be useful in other scenarios.

As a demonstration of useful learned correlations, we implemented a CSN in a classification task using synthetic data. Consider a simplified weather prediction task in which, given noisy observations of temperature and precipitation, a weather station must classify the day as hot or cold and rainy or sunny. In the artificial world, in the last year of weather data, half of the days are rainy and half are sunny, and all rainy days are cold and all sunny days are hot. Cold days have a true temperature uniformly drawn between 0.0 and 0.2 and warm days have a true temperature uniformly drawn between 0.8 and 1.0. Similarly, sunny days have a precipitation value drawn uniformly between 0.0 and 0.2 and rainy days have precipitation values are drawn uniformly between 0.8 and 1.0. Observations of temperature and precipitation are corrupted by zero-mean gaussian noise with $\sigma = 0.05$. Given noisy observations of temperature and precipitation, a model's task is to predict binary labels for whether the day is hot or cold and rainy or sunny.

Numerous causal paths could explain observational data recorded from this environment in which hot days are sunny and cold days are rainy: rain could cause cold weather, some latent factor like atmospheric pressure could affect both precipitation and temperature, etc.. Trained simply from observational data, models are unable to learn the right causal relationship between these variables.

Unlike traditional neural networks, however, CSNs allow humans to encode desirable causal relationships. We designed a CSN with two concept subspaces (for temperature and precipitation), each with two prototypes. We then penalized $(a(Q_{rs}, Q_{hc}) - \sqrt{\rho^*})^2$; that is, we set an intercept of $\sqrt{\rho^*}$ for the alignment loss between the two subspaces for rainy and sunny ($rs$) and hot and cold ($hc$). (We used $\sqrt{\rho*}$ as the notation for setting desired alignment for reasons that will become apparent in the next paragraph.)

In our experiments, we sought to identify if CSNs could learn the desired causal relationship between temperature and precipitation. We did so by setting $\sqrt{\rho^*}$ to some value, training CSNs using standard losses, and then measuring if the $\rho$ metric we calculated from the trained CSNs matched $\rho^*$.

We trained 10 CSNs with latent dimension 2 with $\sqrt{\rho^*} = 0.5$. This corresponds to a cosine value of 0.7, or about 45 degrees. This is intuitively interpreted as meaning that for every percentage increase in likelihood in the weather being sunny, the likelihood of it being warm should increase by 0.7 percent.

As desired, the trained CSNs had a mean $\rho$ value of 0.72 (standard deviation 0.14). An example latent space from one such CSN is shown in Figure 7: the two subspaces are arranged at roughly 45 degrees. Moving an embedding of a cold and rainy day to increase the likelihood of it begin sunny by 1% increases the predicted likelihood of it being warm by $0.7\%$. This demonstrates that we were able to train CSNs to learn the causal relationship we wished for. Lastly, we note that we repeated these experiments with other $\sqrt{\rho^*}$ and found similar findings, and if we did not include alignment training loss, CSNs learned arbitrary concept relationships.

## F  CSN Implementations

In this section, we include details necessary for replication of experimental results that did not fit in the main paper. [1]

In all experiments, we used random seeds ranging from 0 to the number of trials used for that experiment. Although CSNs support several prototypes per class (e.g., 2 prototypes for the digit 0, 2 prototypes for the digit 1, etc.), unless otherwise noted, we used an equal number of prototypes and classes.

In the German fairness experiments, we trained for 30 epochs, with batch size 32, with classification loss weights of 1, alignment loss of 100 between the two subspaces, and KL lossses of 0.5. In the Adult fairness experiments, we trained for 20 epochs, with batch size 128, with classification loss weights of $[1, 0.1]$ for $y$ and $s$, respectively, alignment loss of 100 between the two subspaces, and KL losses of 0.1. For both fair classification tasks, the CSN encoders comprised 2 fully-connected layers with ReLu activations and hidden dimension 128, outputting into a 32-dimensional latent

---

[1]Anonymized code is available at https://anonymous.4open.science/r/csn-8470/

space, and the decoder comprised 2, 128-dimensional ReLu layers followed by a sigmoid layer. The whole model was trained with an Adam optimizer with default parameters.

For the digit and fashion hierarchical classification tasks, CSNs were trained for 20 epochs with batch size 128. CSNs for both datastes used identical architectures to the networks created for the fair classification tasks.

For the CIFAR100 hierarchical classification task, we built upon a ResNet18 backbone, as done by Garnot & Landrieu (2020). The encoder consisted of a ResNet encoder, pre-trained on imagenet, followed by two fully-connected layers with hidden dimension 4096, feeding into a latent space of dimension 100. The decoder (and decoding training loss) was removed in this domain to reduce training time. The network was trained using an SGD optimizer with learning rate 0.001 and momentum 0.9, with batch size 32. Training terminated after 60 epochs or due to early stopping, with a patience of 10 epochs. Classification loss weights were set to $[1, 5]$ for classifying the high- and low-level categories, respectively. KL losses were set to 0; alignment loss was set to $-10$ to encourage parallel subspaces.

For the fair and hierarchical classification task with bolts, CSNs used the same architecture as for the fair classification task. Models were trained for 50 epochs, with batch size 256. All classification loss weights were set to 1; alignment loss between bolts and LR was set to $-10$ and between bolts and participant was set to 100. KL losses were set to 2.

## G  FAIR CLASSIFICATION BASELINES

In Section 4.1, we compared CSNs to several fair classification baselines on the standard German and Adult datasets. Although the datasets are standard in the literature, there are a wide variety of fairness metrics, only a subset of which each method has published. Therefore, we implemented each fair classification baseline and recorded all metrics of interest for each method. In this section, we demonstrated the soundness of our implementations by comparing to published metrics. Implementations of our baselines are available here: https://anonymous.4open.science/status/fair-baselines-44A7.

Tables 8 and 9 report the recorded metrics for each fairness technique. Values reported in prior literature are included in the table using the technique's name (e.g., the second row of Table 8, labeled 'Adv.' includes the values reported by Xie et al. (2017)). Values that we measured using our implementations of each technique are marked with asterisks. We report means and standard errors for our implementations for each method and compare to the metrics that prior methods published for each dataset.

The bottom halves of Tables 8 and 9 are separated from the top halves to indicate modified datasets. The Wass. DB baseline used the German and Adult datasets but treated protected fields differently (e.g., by creating binary age labels at the cutoff age of 30 instead of 25, as all other techniques did). We therefore evaluated CSNs and our own implementations of Wass. DB on these different datasets as well and reported them below the horizontal line.

Interestingly, while we were able to recreate the Wass. DB results on these modified datasets, the technique, when applied to the standard datasets, demonstrated better fairness than most techniques but worse $y$ Acc.. (We repeated the hyperparameter sweeps reported by Jiang et al. (2020) and used the best results.) We attribute the low $y$ Acc. to the fact that Jiang et al. (2020) call for a linear model as opposed to deeper neural nets used by other approaches. When using the datasets suggested by Wass. DB, we reproduced their published results, as did CSNs trained on the same datasets. We note, however, that predictors on this dataset are of limited use, as both Wass. DB and CSNs fail to outperform random classification accuracy.

As a whole, Tables 8 and 9 give us confidence in our implementations of the fairness baselines. Our implementations were able to match or exceed metrics reported from prior art. This suggests that our underlying implementations were correct and that new metrics we gathered on them were valid.

Table 8: Adult dataset fairness results for CSNs and baselines. Results from our baselines are indicated with asterisks. Means (standard deviation) over 20 trials given.

| Model | $y$ Acc. | $s$ Acc. | D.I. | DD-0.5 |
|---|---|---|---|---|
| $CSN_1$ | 0.85 (0.00) | 0.67 (0.00) | 0.83 (0.01) | 0.16 (0.01) |
| Adv. | 0.84 | 0.67 | | |
| Adv.* | 0.85 (0.00) | 0.67 (0.01) | 0.87 (0.01) | 0.16 (0.01) |
| VFAE | 0.81 | 0.67 | | |
| VFAE* | 0.85 (0.01) | 0.70 (0.01) | 0.82 (0.02) | 0.17 (0.01) |
| FR Train | 0.82 | | 0.83 | |
| FR Train* | 0.85 (0.00) | 0.67 (0.00) | 0.83 (0.01) | 0.16 (0.01) |
| Wass. DB* | 0.81 (0.00) | 0.67 (0.00) | 0.92 (0.01) | 0.08 (0.01) |
| Random | 0.76 | 0.67 | | |
| $CSN_2$ | 0.76 (0.00) | 0.77 (0.00) | 0.98 (0.01) | 0.03 (0.00) |
| Wass. DB | 0.76 | | | 0.01 |
| Wass. DB* | 0.76 (0.00) | 0.77 (0.00) | 1.00 (0.01) | 0.01 (0.01) |
| Random | 0.76 | 0.77 | | |

Table 9: German dataset fairness results for CSNs and baselines. Results from our baselines are indicated with asterisks. Means (standard deviation) over 20 trials given.

| Model | $y$ Acc. | $s$ Acc. | D.I. | DD-0.5 |
|---|---|---|---|---|
| $CSN_1$ | 0.73 (0.02) | 0.81 (0.02) | 0.70 (0.12) | 0.10 (0.09) |
| Adv. | 0.74 | 0.81 | | |
| Adv.* | 0.73 (0.03) | 0.81 (0.02) | 0.63 (0.10) | 0.10 (0.08) |
| VFAE | 0.73 | 0.81 | | |
| VFAE* | 0.72 (0.04) | 0.81 (0.04) | 0.47 (0.10) | 0.23 (0.09) |
| FR Train* | 0.72 (0.03) | 0.80 (0.05) | 0.55 (0.15) | 0.16 (0.08) |
| Wass DB* | 0.72 (0.03) | 0.81 (0.05) | 0.33 (0.34) | 0.02 (0.01) |
| Rand | 0.70 | 0.81 | | |
| $CSN_2$ | 0.70 (0.03) | 0.60 (0.02) | 0.65 (0.13) | 0.04 (0.01) |
| Wass. DB | 0.69 | | | 0.00 |
| Wass. DB* | 0.70 (0.02) | 0.60 (0.04) | 0.37 (0.26) | 0.02 (0.01) |
| Rand | 0.70 | 0.60 | | |

