# OpenReview forum: "Prototype Based Classification from Hierarchy to Fairness"
_ICLR.cc/2022/Conference — ICLR 2022 Submitted_

### Official Review · Reviewer_fPSK · 2021-10-20

**Correctness:** 3
**Technical Novelty And Significance:** 3
**Empirical Novelty And Significance:** 3
**Recommendation:** 6
**Confidence:** 4

**Main Review:**

The paper's main strengths are:
* I found it particularly elegant to phrase both hierarchical classification and fairness in the same language, namely that of classification subspaces which are spanned by prototypes.
* The paper connects to a wide range of concepts, namely hierarchical classification, interpretability, and fairness, such that it is of potential interest to a wide range of researchers.
* The paper reports a wide range of experiments; so wide, indeed, that much experimental material had to be pushed to the appendix. I particularly appreciate the analysis of the hierarchies discovered by the prototype network and the comparison to the ground-truth hierarchy via edit distance.
* The paper is clearly written. I, for one, had no problem following along and would feel well equipped to reproduce the reported results.

The paper's main weaknesses are:
* The wide range of concepts discussed result in a certain lack of focus. Fairness, privacy, and causality are all mentioned but only discussed superficially. For fairness, this is particularly dangerous as readers may be mislead to believe that the proposed notion of orthogonality is sufficient for fairness. However, fairness has many meanings (as the paper acknowledges in the appendix) and only some of them are related to the proposed notion of orthogonality. Therefore, I would advise to revise references to fairness, privacy, and causality ind to mention explicitly that only a narrow notion of these terms is implemented by the proposed model.
* The related work fails to mention the historic roots of the prototype concept. I understand that many recent works in prototype networks make the same mistake but I would still advise to not continue it. Prototype-based classification has - to my knowledge - been pioneered by Kohonen in the late 1980s/early 1990s with his work on Learning Vector Quantization (refer to the review by Nova and Estevez, 2014; doi: [10.1007/s00521-013-1535-3](https://doi.org/10.1007/s00521-013-1535-3) ) and has since been extended in many directions, such as metric learning (Schneider et al., 2009, doi:[10.1162/neco.2009.11-08-908](https://doi.org/10.1162/neco.2009.11-08-908)), or probabilistic classification (Seo and Obermayer, 2003, doi: [10.1162/089976603321891819](https://doi.org/10.1162/089976603321891819) ). The latter extension should be of particular interest because the classification scheme is very similar to the one proposed in this paper.
* While the paper reports many different experiments, any single one seems relatively small with few data sets and (for hierarchical classification) few baselines. Further, I could not find information on the hyperparameter selection (e.g. how many prototypes and how strong the regularization strengths lambda were).

Overall, my recommendation is to accept this paper. While the paper could be more focused, make its own contribution and limitations more clearly, and experiments could be extended, I still believe that most flaws could be addressed with minor adjustments and that the core contribution of the paper is interesting enough to a wide range of scholars that publication is warranted.

Nonetheless, I would appreciate if the authors could help me to deepen my understand of the work by responding to two questions:

* I am not fully convinced that the projection into the plane spanned by the prototypes of a classification problem has any effect on the classification itself. If I understand correctly, the paper uses a softmax on the squared distances to all prototypes for classification (which is entirely reasonable). Now, let $D^2$ be the squared distance between a point $z$ and a prototype $p$, let $d^2$ be the squared distance between the projected point $\tilde z$ and the same prototype $p$, and let $h^2$ be the squared distance between $z$ and $\tilde z$. Since the distances form a right-angle triangle, we obtain $d^2 = D^2 - h^2$. This holds for any prototype in the same classification problem. Accordingly, all projected distances within one classification problem are merely the original distance minus a constant offset. This constant offset gets removed by softmax, anyways. So I would assume that the softmax probabilities are the same - no matter whether a point is projected or not.
* Why was the parity hierarchy used as ground truth for MNIST? Garnot et al. use a hierarchy based on visual similarity of the digits (e.g. 3 and 8). Wouldn't that be more natural?

**Summary Of The Paper:**

The present paper proposes a novel architecture for prototype-based classification to support class hierarchies and fairness. In particular, hierarchies are supported by training the model for multiple classification problems jointly, each in its own subspace of the feature space, spanned by the respective prototypes. For fairness, the paper proposes to make the subspace for the classification between subgroups orthogonal to all other subspaces, such that any change in subgroup membership does not influence any other classification. In a series of experiments, the paper evaluates hierarchical classification and fairness separately as well as jointly and demonstrates equal or superior results to a state-of-the-art approach from the literature.

**Summary Of The Review:**

Overall, my recommendation is to accept this paper. While the paper could be more focused, make its own contribution and limitations more clearly, and experiments could be extended, I still believe that most flaws could be addressed with minor adjustments and that the core contribution of the paper is interesting enough to a wide range of scholars that publication is warranted.

---

> ### Author Response · Authors · 2021-11-11
> **Thank you; minor clarifications addressed.**
>
> We thank the reviewer for their feedback: they understood the work very well, and the suggestions will only strengthen the paper. Here, we address general comments and then specific questions.
>
> **General comments:**
>
> We appreciate the fact that, given the breadth of experiments conducted (fairness, hierarchy, fairness + hierarchy, and even single-concept classification in the Appendix), it was difficult to include further results or experiments in this work. Upon acceptance of this paper, it would certainly be interesting to extend the work into more challenging domains.
>
> The notes on prior literature are particularly appreciated. We were unaware of the long history of prototype-based work and will incorporate it into the next version of the paper, hopefully providing references to future readers. We further understand the subtleties of aspects of fair classification and will highlight in our work that CSNs and our metrics only focus on some aspects of fairness.
>
> **Specific questions:**
>
>   1. “Why project before classification?”
>
> The reviewer is correct in noting that taking the softmax before or after projection does not affect probabilities (because of the orthogonal components canceling in the softmax). We will note this fact in the revised paper.
>
> We explained the procedure as projecting and then taking the softmax because that made more intuitive sense to us. (Because the orthogonal component does not matter anyway, one may explicitly remove it.) We also used the projection because projection is important for another aspect of CSNs: enforcing clustering only within a subspace in the PCN loss. If we clustered without taking a projection, the clustering and classification losses would potentially be in conflict.  Overall, we appreciate the reviewer’s comment and will highlight in the text that the softmax may be taken before or after classification for entirely equivalent results.
>
>   2. “Why was parity hierarchy used for MNIST digit?”
>
> Parity was chosen for two reasons, although any hierarchy may be used. For example, we showed that we could use different hierarchies in the CIFAR100 dataset.
>
> For the MNIST digit dataset, we used parity because 1) it was simpler and 2) although Garnot et al. use an intuitive hierarchy based on visual similarities, here we show that CSNs support arbitrary hierarchies. If we used visually-inspired hierarchies, it would be unclear if CSNs merely were leveraging visual feature similarity or the human-guided hierarchy. By using something as artificial as parity, we know that CSNs are learning from the provided hierarchy.
>
> Overall, we thank the reviewer for their time. They clearly understood the work and have provided several useful points for improving the paper, in particular by noting related literature. We welcome further questions as they arise.

---

> > ### Comment · Reviewer_fPSK · 2021-11-11
> > **Re: Thank you; minor clarifications addressed**
> >
> > I thank the authors for their clarifying responses. I have only one further note: In response to reviewer aHhN you mentioned that the lambdas were selected by hand. This should be made explicit in the paper as a limitation, I think.

---

### Official Review · Reviewer_aHhN · 2021-11-02

**Correctness:** 2
**Technical Novelty And Significance:** 2
**Empirical Novelty And Significance:** 2
**Recommendation:** 3
**Confidence:** 3

**Main Review:**

Strength:
- The paper proposed a new prototype-based approach considering the relationship between two concepts (classification tasks) for fairness and hierarchical classification.

Weakness
- The motivation of this paper is not clear to me. It would be helpful to understand the motivation by giving examples of major applications where both fairness and hierarchical classification should be considered. Also, is there any challenge when training a classifier using a regularization term regarding fairness in the existing hierarchy classifier training method?
- It is not clear that why the two subspaces should be orthogonal and parallel for a fair and hierarchical classifier, respectively. Specifically, in section 3.4:
1. For fair classification, what are the two subspaces? is it correct that the two subspaces are for label classification and sensitive attributes classification (e.g., male vs. female), respectively? Then, does the orthogonal relationship between the prototypes for estimating the sensitive attribute and label prediction guarantee the independence of the actual sensitive attribute and label prediction?
2.  “In hierarchical classification, concepts are highly aligned and therefore parallel: the difference
between a toucan and a Dalmatian is similar to the difference between a generic bird and
dog.” --> In this example, why do parallel two concepts imply the difference between a toucan and a Dalmatian can be similar to the difference between a generic bird and dog? I think that the parallelism of two concepts is not related to the different relationships among prototypes. Then It is not clear that why two concepts should be parallel in the hierarchical classification.

Questions:
- How does one train a hierarchical classifier with 3 or more concepts? In hierarchical classification, I think there are at least 3 concepts (e.g., dog vs bird, dog species classification, bird species classification)
- In experiments, How was the parameter lambdas (in equation 2) chosen in the experiments?

Minor feedback:
- Adding a figure describing the proposed architecture will help readers understand the framework.


**Summary Of The Paper:**

This paper proposed a framework (that authors called the concept subspace network) using prototype-based representation controlling the alignment between two subspaces for the purpose of the classifier (fair or hierarch classification).

**Summary Of The Review:**

The motivation of the paper and the intuition of the proposed approach is not clear.

---

> ### Author Response · Authors · 2021-11-11
> **Clarifications on number of subspaces and intuition of parallelism for hierarchy.**
>
> We thank the reviewer for their feedback. In this rebuttal, we provide a general response, followed by specific answers to the reviewer’s questions.
>
> **General Response:**
>
> We seek to correct an apparent misconception: CSNs support any number of concept subspaces and not just between two subspaces.
>
> The alignment loss, calculated between all pairs of subspaces, supports this general framework. Furthermore, we report results of experiments in which CSNs used more than two subspaces: in one of the CIFAR100 hierarchical experiments, we used 5 subspaces for the 5 levels in the hierarchy, and in Section 4.3, we showed how to use both fair and hierarchical classification by guiding the alignment between three subspaces.
>
> **Specific Questions:**
>
>   1. Why is fair orthogonal and hierarchical parallel?
>
> In the paper, we state that fair classification corresponds to orthogonal subspaces and hierarchical classification to parallel subspaces. At the very least, our results support this claim: by switching the alignment loss from positive (for fairness) to negative (for hierarchical), we induce different model behaviors. We even perform an ablation study in Table 3, studying the effect of alignment on fair classification. Lastly, the latent spaces plotted in Figure 6 show how orthogonality and parallelism arise in trained models. Thus, our results support our claim about alignment and fairness vs. hierarchy. We next explore the intuition behind this result.
>
> In fair classification, we have two subspaces, one for the label classification and one for the sensitive attribute. (These details are implied in Appendix F - “...the two subspaces… with classification loss weights…for y and s...” - but will be made more prominent.) Orthogonality of these subspaces enforces some forms of independence but not the desired fairness characteristics. We demonstrate this in Table 3: turning on alignment loss enforced orthogonality and therefore causal independence between concepts (see the “Align” row) but was insufficient to ensure statistical independence (as shown by worse fairness metrics).
>
> In hierarchical classification, we consider an example of dalmatians and toucans. We seek to train a CSN that reflects the notion that the difference between a dalmatian and a toucan is roughly the same as that between generic dogs and birds. If we succeed in training a CSN such that the dalmatian prototype is near the dog prototype, and toucan near bird, we will have such a CSN. Here, we show that parallel subspaces achieve this goal.
>
> A CSN performs classification by projecting a single encoding of an image into the high- and low-level concept subspaces. To correctly classify an image as both a dog and a dalmatian, the projections must fall close to the prototypes for dog and dalmatian. Similarly, to correctly classify a toucan as both a bird and a toucan, the projections into subspaces must fall close to the prototypes for bird and toucan. If we specify that the two concept subspaces are parallel, the only way for a CSN to achieve high classification accuracy is for the bird and toucan prototypes to be close to each other (and similarly for dog and dalmatian). This therefore confirms that parallel subspaces achieve the desired prototype arrangement for hierarchical classification.
>
>   2. "How does one train a hierarchical classifier with 3 or more concepts?”
>
> Each concept merely requires an additional set of prototypes, as we showed with a 5-level classifier for a CIFAR100 (page 8).
>
> The reviewer describes having a different set of prototypes for bird species and dog species. This is certainly possible and is to some extent similar to work done in the Hierarchical Prototype Network, but is a different approach from ours. In CSNs, there is a single subspace per level in the hierarchy. As a result, we can encourage meaningful concept relationships between levels in the hierarchy (e.g., parallelism). It is less clear what a useful relationship between dog species and the prototype for a dog would be. Furthermore, we validated in experiments that our approach outperformed HPNs, indicating that our approach to defining sets of prototypes was more productive.
>
>   3. “How were the lambdas selected?”
>
> The loss weights were chosen by hand. One could perform more extensive hyperparameter searches; doing so should only improve CSNs reported performance. We note that, overall, these lambdas were not highly optimized. In fairness domains, for example, the alignment loss weight was set to 1000 to make the subspaces orthogonal; in hierarchy domains, the loss weight was set to -10 to encourage parallelism. These values were chosen primarily based on their signs and worked almost right away, indicating a likely robustness to choice of lambdas.
>
> We thank the reviewer for further minor comments about including a figure and will do so in an Appendix, for space reasons. We look forward to clarifying any further questions.

---

### Official Review · Reviewer_z2bK · 2021-11-02

**Correctness:** 3
**Technical Novelty And Significance:** 3
**Empirical Novelty And Significance:** 3
**Recommendation:** 6
**Confidence:** 3

**Main Review:**

In this paper, the ideas are quite novel and mostly well presented, and the problem handled is significant.

I have though some questions and some minor comments that I hope will be addressed in the final version:
1. The way in which concept subspaces are defined is not clear to me. In the paper, the authors write: “Given a set of k prototypes in $R^Z$ , the prototypes define a subspace generated by starting at the first prototype, $p_1$, and adding all linear combinations of vector differences from $p_1$ to all other $p_i$; $i \in [2, k]$.” This is not clear to me, and it would be beneficial having a clearer example than the one given in the paper, in which it is not clear why we should obtain the plane $x-y$.
2. Also, in order to get a subspace, do you need the assumption that $k < Z$?
3. In equation (2) the term $PCN(\cdot)$ is just defined as the loss introduced for PCN. Where is it defined?
4. The random baseline seems to achieve very high performance in tables 1 and 2.
5. At page 7 the authors mention a global ordering of the nodes, how was such ordering decided?


Minor comments:
1. $Z$ in the figure 1 instead of $z$
2.  Add upward and downward arrows nearby the metric names to improve readability
3. “Random” and “Rand” in Tables 1 and 2

**Summary Of The Paper:**

The authors propose a novel model — called Concept Subspace Network (CSN) — for both hierarchical and fair classification. The idea behind the network is to use sets of prototypes to define concept subspaces in the latent space defined by the neural network itself. The relationships between the subspaces can be manipulated at training time to enforce concept relationships (i.e., two concept subspaces are orthogonal if the concepts they represent are independent, while they are parallel if the concepts they represent are hierarchically organised).

**Summary Of The Review:**

The paper can be accepted if some clarifications are made

---

> ### Author Response · Authors · 2021-11-11
> **Clarification of specific questions, with mathematical notatioin of linear subspace.**
>
> We thank the reviewer for their comments and careful reading of the paper. They seem to have understood most of the contributions; in this review, we seek to clarify specific questions and note an additional contribution of our work.
>
> **General comments**
> The reviewer noted our contributions in fair and hierarchical classification tasks. We also wish to emphasize a third domain in which CSNs are valuable: in fair and hierarchical classification simultaneously. Unlike any prior art we are aware of, CSNs may simultaneously use hierarchical information and perform fair classification, as demonstrated in Section 4.3.
>
> **Specific Questions**
>
>   1. “The way in which concept subspaces are defined is not clear to me.”
>
> Other reviewers appeared to be confused by subspaces as well, and we welcome feedback on how to better present this mathematical notation.
>
> Here, we work through the intuition of how to define the subspace. We introduce mathematical notation later.
>
> Let us say we have $k$ prototypes, each of which is a point in $R^z$. Let us pick the first prototype, $p_1$, and draw a vector from $p_1$ to all other prototypes in $R^z$. We now have a set of vectors. The concept subspace is the set of all points that can be reached by the linear combination of these vectors, starting at $p_1$. In the example in Figure 1 a, we arbitrarily created prototypes that fell in the $x-y$ plane, so that was the subspace, but in general the prototypes can form higher-dimensional subspaces if there are more prototypes, and the subspaces certainly do not need to align with axes.
>
> The mathematical notation for the subspace is as follows, again assuming $k$ prototypes in $R^z$.
>
> $v_i$ = $p_i - p_1$ for all $i \in [2, k]$
>
> $C =$ \{ $x | x \in R^z$ where $x = p_1 + \sum_i (\lambda_i v_i)$ for $i \in [2, k]$ and all $\lambda_i \in R$\}
>
> That is, we define a vector $v_i$ from $p_1$ to $p_i$ for all $i$.
> We then say that $C$ is the concept subspace, comprising all points $x$ for which $x$ is equal to the sum of $p_1$ and a linear combination of all vectors $v_i$. If we have one vector, that describes a line; if there are two vectors, that describes a plane.
>
> We appreciate the feedback that the quick explanation of prototypes defining a subspace was insufficient. Does the proposed explanation clarify the issue?
>
>   2. “To get a subspace, do you need the assumption that k < Z?”
>
> This is a good question. In order for the concept subspace to be a strict subspace of the overall space $R^Z$, yes, it is necessary for $k$ to be less than $Z$. However, if $k >= Z$, that merely creates the scenario in which the concept subspace spans the full latent space. In that case, projecting into the subspace has no effect. In fact, we used induced such a case in the CIFAR100 experiments: the finest-level classification used 100 prototypes but the latent dimension was also 100, as noted in Appendix F. Thus, although in most of our experiments, $k < Z$, in general we support $k >= Z$ as well.
>
>   3. “Where is the term PCN(.) defined?”
>
> Due to space constraints, PCN merely referred to “the loss introduced for the PCN, encouraging classification accuracy and the clustering of encodings around prototypes.” We refer the reviewer to Equation 7 in the original PCN paper. We welcome suggestions for how to include more details, given space constraints.
>
>   4. “High random baseline performance in Tables 1 and 2”
>
> The reviewer’s assessment is correct: random baseline classification often performs close to, but below, trained classifiers. This is merely a symptom of the datasets and has been reported in prior art as well.
>
>   5. “Global ordering of the nodes”
>
> The ordering was determined by going from high to low classification (e.g., even vs. odd to specific digit), with orderings within a level of abstraction determined by numerical ordering of the label (e.g., 0 then 1 then 2 for digit classification). The important aspect was that edges pointed from higher-level categories to lower-level categories, which we also did for constructing the ground truth trees.
>
>   6. Minor comments
>
> We appreciate the reviewer’s detailed comments and will make the suggested improvements.
>
>
> Overall, we again thank the reviewer for their feedback and look forward to clarifying any further questions.

---

> > ### Comment · Reviewer_z2bK · 2021-11-12
> > **Clarifications ok - please include them in the paper**
> >
> > I thank the authors for their comments. The paper is much clearer now.
> >
> > I strongly believe that the new mathematical notation for concept subspaces needs to be added to the paper.
> >
> > Also, I ask the authors to clarify that the choice of the prototypes in Fig. 1(a) was arbitrary and that, as a result, they got the plane $x-y$. It would probably be a good idea to create a separate latex environment "example" with all the details, in this way the readers can easily understand the content of your paper.
> >
> > Regarding the PCN loss, a paper should be self-contained, and hence the authors should add that in their papers.
> >
> > Finally, I have a question, can this method be extended to hierarchical multi-label classification problems? (see e.g., [1], [2])
> >
> > [1] J. Wehrmann, R. Cerri, and R. C. Barros.  Hierarchical multi-label classification networks. In Proceedings of ICML, 2018.
> >
> > [2] E. Giunchiglia and T. Lukasiewicz. Coherent hierarchical multi-label classification networks. In Proceedings of NeurIPS, 2020.

---

> > > ### Author Response · Authors · 2021-11-12
> > > **Answering and asking questions about hierarchcial multi-label classification**
> > >
> > > Thank you for the quick response. It is very useful knowing what changes help, including the mathematical notation for subset, explaining the arbitrary nature of the x-y plane in the example (currently described only as "corresponding in this case to the $x-y$ plane"), and including details of the PCN loss.
> > >
> > > We thank the reviewer for the hierarchical multi-label classification literature. We were unaware of this class of problem. The current CSN formulation uses a softmax over distances to prototypes, implying just  a single label per level. However, one could certainly change that activation to be a sigmoid function using the distance to each prototype. That would therefore output a binary prediction for each label, as described in Giunchiglia and Lukasiewicz Section 2. We look forward to running experiments in such domains. We do ask the reviewer for advice, though: given that the paper already considers 3 types of problems (fair, hierarchical, and fair and hierarchical) and already relegates single-concept classification problems to an appendix, does the reviewer think hierarchical multi-label classification is more worthwhile than the currently-presented problems? Because we are new to this domain, we welcome advice.
> > >
> > > Thank you again.

---

> > > > ### Comment · Reviewer_z2bK · 2021-11-12
> > > > **Hierarchical multi-label classification answers**
> > > >
> > > > Personally, I believe it would make the paper even stronger, since hierarchical classification can be seen as a particular case of hierarchical multi-label classification (HMC).
> > > >
> > > > Further, these hierarchies can be quite deep (up to 13 levels), thus the authors could really show the power of their model here.
> > > >
> > > > Overall, (if I were the authors) I would surely add a paragraph in the hierarchical classification section in which I'd explain how the model can be easily generalised for the HMC setting, and, if possible, I would take couple of datasets from the HMC world and I would show what happens to the concept subspaces in that case.

---

### Official Review · Reviewer_BhqD · 2021-11-04

**Correctness:** 2
**Technical Novelty And Significance:** 2
**Empirical Novelty And Significance:** 2
**Recommendation:** 3
**Confidence:** 3

**Main Review:**

The paper attempts interesting problems but lacks on 2 major fronts: (1) It is not clear what the improvement over the existing work is, and if it is significant enough to merit acceptance at ICLR (2) the writing needs a lot of work to bring out motivation for different choices.

About the first point, the main contribution seems to lie in section 3.1, but most of the machinery here seems to be borrowed from the PCN work of Li et al. 2018. The additional contribution seems to be the concept subspace projection (if I understood correctly), whose motivation is not explained very well, and the addition of the alignment term in Eq. 1. The paper does not explain what is the additional value added by these terms over PCN.

Continuing from the previous point, the paper is very hard to understand. In the second paragraph of Section 3.1, there is some departure from PCN where some combinations of $p1$ and other prototypes are taken. The process is defined in a very handwavy manner and not clear what it is the formal mathematical operation performed here. How is this different from PCN and why was this step needed? Talking about digits and concept subspaces, do we have as many concept subspaces as the number of classes? If not, how is this number picked?

Moving on to the third paragraph of 3.1, the first few lines seem to be quite similar to PCN. However, at some point, a probability distribution is mentioned in connection with traditional softmax probabilities, but then yet another probability distribution is mentioned. It is not clear what the second distribution does. In the absence of formal equations, it is very difficult to understand what each component does. I would highly recommend describing each operation formally (in a sequential manner) and also adding a visualization like Figure 1 in the PCN paper to clearly convey the idea to the reader.

Fourth paragraph of 3.1 mentions two differences from PCN. Again, it is not clear what each of the differences achieves. Moreover, Figure 1 is neither described well in the main text nor in the caption, leaving the reader puzzled over what is happening in the figure. Instead of the regular autoencoder, a variational autoencoder is used, but again, the motivation is not clear. Other important details like the text above Equation 1, and the usage of KL diveregnce regularization term are skimmed over very quickly. The details of hierarchical classification setup in 3.4 are also glossed over quickly. The same happens in 4.2. For instance, what is meant by "adopting the conditional probability training loss introduced by Hase et al"?

**Summary Of The Paper:**

The paper builds on prior work on prototypical classification networks (more specifically, the work of Li et al. 2018) and additionally tries to include criteria such as orthogonality to enable applications such as fair classification. An application to hierarchical networks is also described though the details are very hard to understand. Experiments show that the resulting models are able to achieve reasonable fairness accuracy tradeoffs.

**Summary Of The Review:**

The contributions are not clear and the writing needs major work.

---

> ### Author Response · Authors · 2021-11-11
> **Clarifying technical contributions beyond fairness and small changes to PCN**
>
> We thank the reviewer for their consideration of our work. In this review, we hope to illustrate the technical merits of this contribution, therefore motivating acceptance to the conference. After a summary, we address specific questions at the end of this rebuttal.
>
> In this work, we define the concept subspace network (CSN), a neural network architecture. In a CSN, we define a set of prototypes for each classification task. For example, in fair classification, we have two sets of prototypes: one for the protected class and one for the predicted class. These separate sets of prototypes define separate subspaces. This is illustrated in Figure 1 b, c, and d, for example: there are two sets of prototypes, and one set defines the gray subspace and the other set defines the blue subspace. Thus, answering the reviewer’s question, “Do we have as many concept subspaces as the number of classes?” we state: there are as many concept subspaces as classification tasks.
>
> The reviewer has carefully analyzed Section 3.1, but we argue that the biggest technical contribution of our work is described in Section 3.2. In that section, we describe how a single CSN may define several concept subspaces and the alignment between them: this is novel compared to prior art.
>
> The technical contributions of our work are confirmed in a series of experiments in different domains. The reviewer correctly notes that CSNs achieve a reasonable fairness-accuracy tradeoff. We also note that we performed experiments in two other types of domains beyond fairness. First, CSNs match or in some cases exceed state of the art performance in hierarchical classification domains (Section 4.2). In these tasks, the CSN must classify an input at each label of a hierarchy; we showed that CSNs achieved high classification accuracies and also, when making mistakes, on average made “less costly” mistakes than prior art. Second, in Section 4.3, we identify a domain in which fair and hierarchical classification are needed simultaneously, and we demonstrate how CSNs can do so, unlike any prior art we are aware of. We therefore urge the reviewer to reconsider their assessment of our experiments. We do not merely match fairness state of the art: we also show how CSNs can be brought to bear on a range of problems that prior methods only solved separately or not at all.
>
> **Specific Questions**
>   1. “The main contribution seems to lie in section 3.1”
>
> Although there are some technical changes introduced in Section 3.1, the main contributions are introduced in the later subsections of Section 3. In 3.2, we introduce multiple concept subspaces and notions of their alignment; in 3.3, we show how one may control alignment via a training loss; in 3.4, we show how fair and hierarchical classification problems may be characterized as different extremes of concept alignment. We therefore emphasize that the key contribution of our work is that of concept alignment, which is introduced in section 3.2 rather than 3.1.
>
>   2. “What is the additional value added by these terms over PCN”
>
> The key change of CSNs over PCNs is using multiple concept subspaces. (We do make some small changes relative to PCNs for single classification tasks, as detailed in 3.1, but that is not the key idea.) PCNs only define one set of prototypes and can only be used for a single classification task. Conversely, CSNs can perform multiple classification tasks simultaneously because CSNs have multiple sets of prototypes, one for each classification task, which in turn supports the various classification tasks used in our experiments.
>
> For specific analysis of the importance of the alignment term in Equation 1, we direct the reviewer to the ablation study in Table 3 (where we show that alignment is necessary for fair classification). We also note that the key difference between the fair and hierarchical classification experiments was simply a change in sign in the alignment loss: therefore, the alignment term was the critical component in enabling the desired behaviors.
>
>   3. “What is the formal mathematical operation performed here?”
>
> We assume the reviewer is asking about how prototypes were used in classification (e.g., “some combinations of p1 and other prototypes are taken.”).  We are not combining prototypes: we are merely defining a linear subspace spanned by the prototypes. Reviewer z2bK similarly asked for mathematical notation for the subspace: we direct the reviewer to our response to them (Specific Question 1).
>
>   4. What is meant by "adopting the conditional probability training loss introduced by Hase et al"?
>
> That sentence describes how HPNs (prior art by Hase et al.) differ from CSNs and are trained with a different training loss. Because of space constraints, we left out the exact training loss. If the reviewer seeks further details, we are happy to elaborate.
>
>
> We look forward to answering any further questions the reviewer has.

---

### Decision · Program_Chairs · 2022-01-20

**Decision:**

Reject

**Comment:**

This paper extends prototypical classification networks to handle class hierarchies and fairness. New neural architecture is proposed and experimental results in support of it are presented.

Unfortunately, reviewers found that paper in its current for is not sufficiently strong to be accepted at ICLR. Authors have made a significant attempt to clarify and improve the paper in their response. However, reviewers believe that contributions and motivation can be clarified further. We encourage authors to improve their work according to the specific suggestions made by the reviewers and resubmit.